# More Than Generation: Unifying Generation and Depth Estimation via Text-to-Image Diffusion Models

**Hongkai Lin**   **Dingkang Liang**   **Mingyang Du**   **Xin Zhou**   **Xiang Bai**[✉]

Huazhong University of Science and Technology

{hklin,dkliang,xbai}@hust.edu.cn

Project code: https://github.com/H-EmbodVis/MERGE

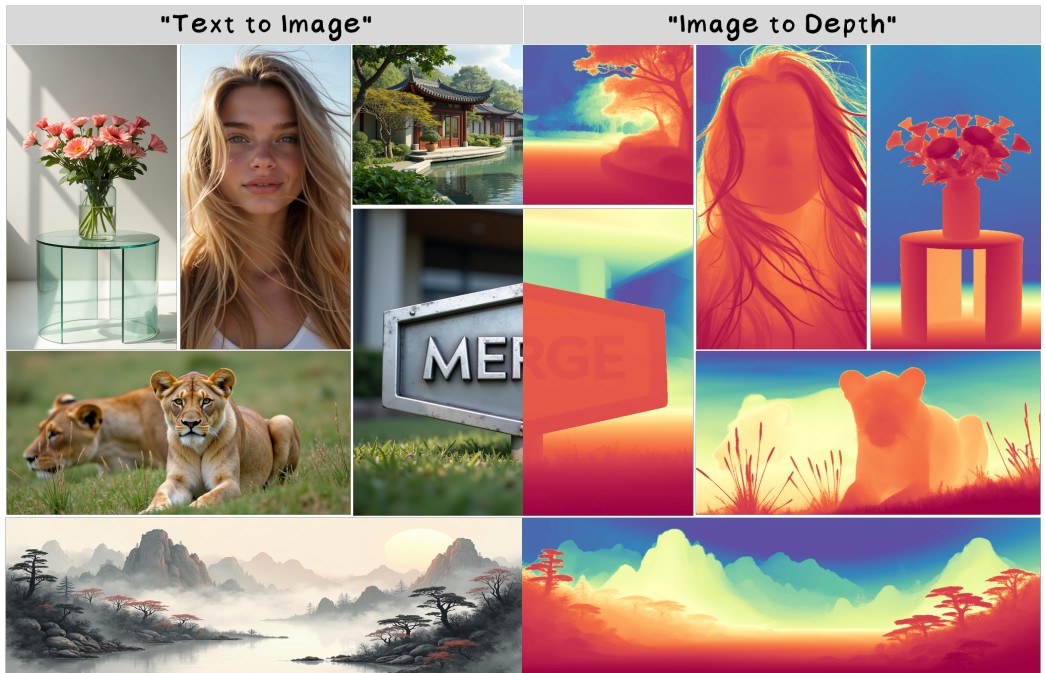

Figure 1: We present MERGE, a simple unified diffusion model for image generation and depth estimation. Its core lies in leveraging streamlined converters and rich visual prior stored in generative image models. Our model, derived from fixed image generation models and fine-tuned pluggable converters with synthetic data, expands powerful zero-shot depth estimation capability.

## Abstract

Generative depth estimation methods leverage the rich visual priors stored in pre-trained text-to-image diffusion models, demonstrating astonishing zero-shot capability. However, parameter updates during training lead to catastrophic degradation in the image generation capability of the pre-trained model. We introduce MERGE, a unified model for image generation and depth estimation, starting from a fixed pre-trained text-to-image model. MERGE demonstrates that the pre-trained text-to-image model can do more than image generation, but also expand to depth estimation effortlessly. Specifically, MERGE introduces a play-and-plug framework that enables seamless switching between image generation and depth estimation

[✉] Corresponding author.

39th Conference on Neural Information Processing Systems (NeurIPS 2025).

modes through simple and pluggable converters. Meanwhile, we propose a Group Reuse Mechanism to encourage parameter reuse and improve the utilization of the additional learnable parameters. MERGE unleashes the powerful depth estimation capability of the pre-trained text-to-image model while preserving its original image generation ability. Compared to other unified models for image generation and depth estimation, MERGE achieves state-of-the-art performance across multiple depth estimation benchmarks.

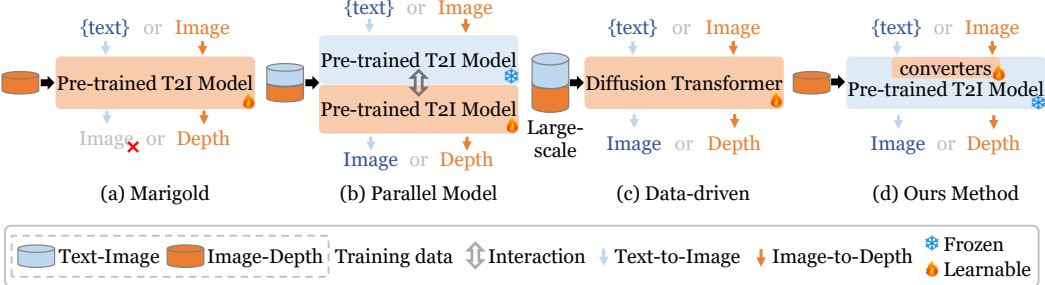

Figure 2: The comparison between existing methods and ours shows that, unlike previous works, our method requires only a few additional parameters to unleash its powerful depth estimation capability without compromising its inherent T2I generation ability.

## 1 Introduction

Diffusion models [12], especially text-to-image (T2I) models [1, 19, 38], have made astonishing progress in the quality of generated images. Meanwhile, emerging generative depth estimation research [10, 11, 16] reveals the latent potential for depth estimation tasks of advanced T2I models. Exploring a framework bridging image generation and depth estimation is an essential step toward building a Unified Model used for generation and visual perception.

Marigold [16], a pioneer of generative depth estimation, shows a powerful zero-shot depth estimation method leveraging pre-trained T2I diffusion models, as shown in Fig. 2(a). Even though the irreversible degradation of its original generation capability due to parameter updates during training, the paradigm still makes it possible to unify image generation and perception within diffusion models. In contrast, JointNet [52] and UniCon [21] present an alternative strategy by utilizing a parallel dual-model interaction architecture to unify generation and depth estimation tasks, as shown in Fig. 2(b). The latest work, OneDiffusion [20], introduces a data-driven solution, as shown in Fig. 2(c), to train a unified generation and perception model from scratch with massive multitask data (100M).

Despite remarkable progress, such approaches may be considered inefficient or resource-intensive solutions, potentially suboptimal choices. Considering rich visual prior is stored in advanced T2I models, a natural question arises: *Can T2I models effortlessly expand depth estimation capability without degrading their image generation capability?* This paper explores a unified diffusion model without bells and whistles for image generation and depth estimation tasks, starting from a fixed T2I model. Our method, referred to as **MERGE** (as shown in Fig. 2(d)), demonstrates that T2I models can do **M**ore than gen**ER**ate ima**GE**s. With a few simple converters, they can effortlessly unleash the powerful depth estimation capability.

To enable depth estimation ability in a fixed T2I diffusion model, we present a play-and-plug framework without bells and whistles. Before each transformer layer of the pre-trained T2I diffusion transformer (DiT), hereafter referred to as the T2I block, we introduce an identical, learnable T2I block as a converter. This converter transforms the latent features originally tailored for the T2I task into features suitable for the depth estimation task. This straightforward play-and-plug design enables seamless switching between the original image generation and depth estimation models by skipping or running these converters. Meanwhile, considering the similarity of output features between pre-trained T2I blocks, we propose a **G**roup **RE**use (GRE) mechanism to improve the utilization of the converter. Specifically, pre-trained T2I blocks are divided into several groups, and a shared within-group converter is used to effortlessly transform the T2I model into a generative

depth estimation model. During inference, if the text-to-image generation capability is needed, these additional converters can be easily skipped, restoring the depth estimation model to the T2I model. In addition, some empirical studies further simplify the converters with virtually no impact on MERGE's depth estimation performance. Combining these simple, flexible, and pluggable converters with the group reuse mechanism enables MERGE to unify image generation and depth estimation with no difficulty. Remarkably, MERGE presents a unified model starting from a fixed T2I model, requiring only a few additional parameters and fine-tuning on depth estimation datasets.

We conduct comprehensive evaluations of MERGE on established depth estimation benchmarks (NYUv2 [40], ScanNet [6], and DIODE [43]). The results show that MERGE achieves state-of-the-art performance across multiple metrics and benchmarks while introducing merely 12% additional trainable parameters. Especially on the NYUv2 benchmark, MERGE achieves 5.9 A.Rel and 95.4% $\delta1$, outperforming OneDiffusion, which is trained from scratch with massive data.

MERGE demonstrates a simple and effective approach to enabling pre-trained T2I diffusion models with depth estimation capability while preserving their inherent T2I generative ability. We hope our work provides valuable insights into the construction of unified models for generation and perception, while offering a cost-effective solution ($\sim 12\%$ additional parameters) for extending the capabilities of pre-trained image generation models. The main contributions of this work are as follows: **1)** We explore how to unleash the depth estimation capability from fixed pre-trained T2I models while preserving their original generation ability and propose a simple method without bells and whistles called MERGE. The key lies in a flexible play-and-plug framework that allows seamless switching between image generation and depth estimation modes. **2)** We present a streamlined converter that effortlessly transforms features suited for image generation into depth estimation. Considering the feature similarity between different layers in pre-trained T2I models, we propose a Group Reuse Mechanism, which encourages converter reuse to improve parameter efficiency.

## 2 Related Work

**Text-to-Image Diffusion Model.** Recently, text-to-image (T2I) generation models [1, 30, 39] based on the diffusion principle [12] make significant progress in image quality, diversity, and efficiency. Stable Diffusion [38], as the pioneering latent diffusion model, compresses the T2I generation process into the latent space using VAE [17], greatly improving computational efficiency and the quality of generated images. SDXL [32] introduces dual text encoders [15, 33] and fully upgrades Stable Diffusion, further enhancing the quality and resolution of generated images. VQ-Diffusion [9] proposes a vector quantized diffusion model by replacing VAE with VQ-VAE [41], addressing the unidirectional bias problem. Unlike these denoising diffusion models based on UNet architectures, PixArt [4, 5] introduces a novel text-to-image paradigm built upon Diffusion Transformers (DiT) [31]. By integrating vision-language models [27] to construct high-quality text-image pairs data, PixArt achieves performance comparable to advanced works at significantly reduced computational cost. MMDiT, an innovative Multimodal Diffusion Transformer architecture, is introduced by Stable Diffusion 3 [7] and demonstrates excellent image generation capability. The state-of-the-art method, FLUX [19], builds a larger T2I diffusion model by mixing MMDiT with the Single-DiT, presenting stunning image generation capability. Distinguishing these excellent T2I works, this paper focuses on exploring how to unleash the depth estimation capability of pre-trained T2I models (specifically those transformer-based architectures) while preserving their inherent T2I generation ability.

**Diffusion-based Depth Estimation Model.** Unlike discriminative perception methods [23, 24, 36, 45, 46, 48] and these works [22, 44, 47, 54] that treat T2I models as feature extractors, emerging research [14, 16, 25] reveals the compelling potential of pre-trained T2I models as generative depth estimation models. GeoWizard [8] leverages a geometry switcher inspired by Wonder3D [29] to extend a single stable diffusion model to produce depth and normal results. Meanwhile, a simple yet effective scene distribution decoupler strategy proposed in this work boosts the capture of 3D geometry. DepthFM [10] is the pioneer of combining Flow Matching [26] to explore generative monocular depth estimation, significantly improving inference speed while maintaining accuracy. BetterDepth [53] is a straightforward and powerful depth estimation framework that refines the results of generative depth estimation by using the output of a foundation depth estimation model [48, 49] as a prior. Lotus [11] introduces a tuning strategy called detail preserver that achieves more accurate and fine-grained predictions, especially on reflective objects. JointNet [52] and UniCon [21] propose a unified architecture for image generation and depth estimation, leveraging parallel dual diffusion

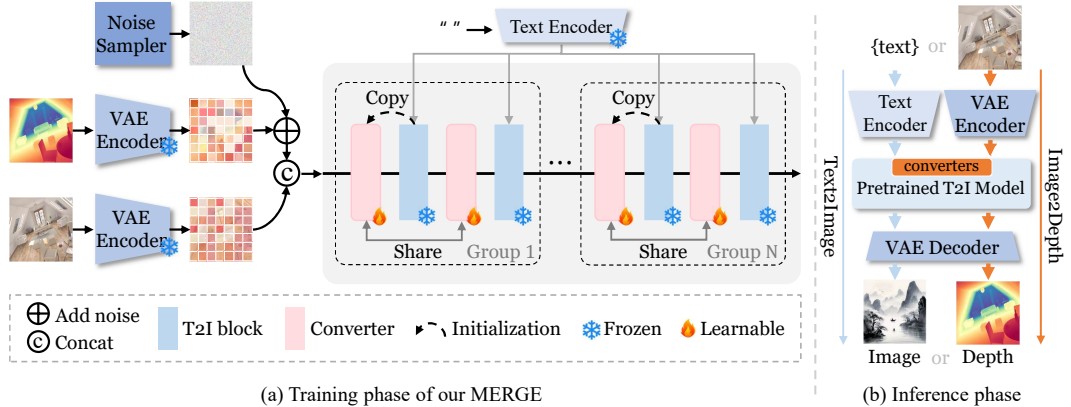

(a) Training phase of our MERGE          (b) Inference phase

Figure 3: The pipeline of MERGE. Starting from the fixed DiT-based text-to-image (T2I) model, where transformer layers (hereafter referred to as T2I blocks) are divided into different groups. A shared and learnable converter is inserted before each T2I block within a group, transforming it into a depth estimation model. It can be reverted to the original T2I model by skipping these converters.

models with feature interaction operations between them. The latest work, OneDiffusion [20], presents a massive data-driven unified model for image generation and depth estimation. In contrast to existing full-parameter fine-tuning methods, which catastrophically degrade the image generation ability of pre-trained T2I models, or resource-intensive approaches, we explore a method that preserves the inherent generation capability of the T2I model while seamlessly transforming it into a generative depth estimation model.

## 3 Preliminaries

In this section, we review latent text-to-image diffusion models, with Stable Diffusion [38] serving as a representative of the latent diffusion paradigm.

Diffusion Models (DMs) [12] establish advanced text-to-image generation frameworks renowned for synthesizing photorealistic images through iterative denoising. DMs learn to reconstruct complex data distributions by approximating the reverse of a predefined diffusion process. Denoting $z_t$ as the random variable at the $t$-th timestep, the diffusion process is modeled as a Markov Chain:

$$z_t \sim \mathcal{N}(\sqrt{\alpha_t}z_{t-1}, (1-\alpha_t)\boldsymbol{I}), \tag{1}$$

where $\alpha_t$ is a fixed coefficient predefined in the noise schedule, and $\boldsymbol{I}$ refers to the identity matrix. A prominent variant, the Latent Diffusion Model (LDM) [38], innovatively shifts the diffusion process of standard DMs into a latent space. This transition notably decreases computational costs while preserving the generative quality and flexibility of the original model. The resulting efficiency gain primarily arises from the reduced dimensionality of the latent space, which allows for lower training costs without compromising the model's generative capability.

Stable Diffusion, an exemplary implementation of LDM, comprises an AutoEncoder [41] and a latent denoising model. The AutoEncoder $\varepsilon$ is designed to learn a latent space that is perceptually equivalent to the image space. Meanwhile, the LDM $\epsilon_\theta$ is parameterized as a denoising model with the multimodal feature interaction module and trained on a large-scale dataset of text-image pairs via:

$$\mathcal{L}_{LDM} := \mathbb{E}_{\varepsilon(x),y,\epsilon\sim N(0,1),t}[\|\epsilon - \epsilon_\theta(z_t, t, \tau_\theta(y))\|_2^2], \tag{2}$$

where $\epsilon$ is the target noise, and $x$ is an RGB image. $\tau_\theta$ and $y$ are the pre-trained text encoder (e.g., CLIP [33], T5 [34]) and text prompts, respectively. This equation represents the mean-squared error (MSE) between the target noise $\epsilon$ and the noise predicted by the model, encapsulating the core learning mechanism of the latent diffusion model. Some of the latest text-to-image works (e.g., Stable Diffusion 3.5 [7], FLUX [19]) attempt to introduce more advanced flow matching [26] as an optimization objective, as it can more directly establish a mapping between the noise and data distributions, further improving image generation quality and efficiency.

# 4 Our MERGE

To unleash the potential depth estimation capability of pre-trained text-to-image (T2I) models while retaining their inherent image generation capability. We present MERGE, as shown in Fig. 3(a), which requires only simple modifications to the pre-trained T2I model that can effortlessly unify image generation and depth estimation. The core of MERGE lies in a play-and-plug framework and the Group Reuse Mechanism without bells and whistles, designed for unified image generation and depth estimation. In addition, we present important empirical investigations, which significantly reduce the number of learnable parameters while keeping depth estimation capability virtually unaffected.

## 4.1 Play-and-Plug Framework

The T2I diffusion model learns the training data distribution to generate infinite data. Among these infinite pixel-level combinations that form images, there may exist underlying information highly correlated with the depth estimation modality. However, due to weak prompts and the sparse representation of depth information within the overall data distribution, extracting this latent feature of depth estimation modality from a fixed pre-trained T2I model is challenging.

In contrast to previous works [10, 11, 16] on fine-tuning the denoising models with full parameters, which catastrophically disrupt the inherent generation ability, we introduce play-and-plug, learnable converters before each T2I block in the pre-trained T2I model to guide and unleash its potential depth estimation capability. This play-and-plug converter design more naturally leverages the visual priors stored in the pre-trained T2I model, presenting highly comparable or even better depth estimation capability than full-parameter fine-tuning with fewer training parameters.

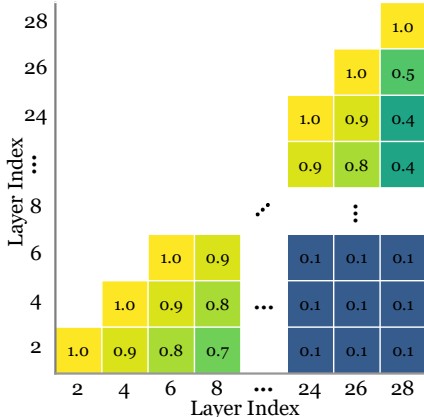

Features similarity between T2I blocks

Figure 4: The cosine similarity between the output features of different T2I blocks within the PixArt [5] model.

This straightforward strategy, without bells and whistles, demonstrates remarkable depth estimation capability and preserves the inherent image generation capability of the pre-trained T2I model, which is an aspect that previous works do not explore.

## 4.2 Group Reuse Mechanism

The solution of copying a learnable block before each pre-trained T2I block as a converter is simple yet effective. However, this method, which can be seen as directly coupling two models, does not efficiently leverage the powerful visual knowledge of the pre-trained T2I model to seamlessly guide its latent depth estimation capability, making it an inefficient approach. Interestingly, we observe that the closer two T2I blocks in the pre-trained T2I model are more similar in their output features in most cases, as shown in Fig. 4. Considering this observation, we propose a Group Reuse Mechanism (GRE) to encourage converter reuse and improve parameter utilization. Specifically, the T2I block in the pre-trained denoising model is divided into different groups, as shown in Fig. 3(a). The T2I block in a group shares a converter, which transforms the latent features suitable for the image generation task into the depth estimation task. Since GRE considers the feature similarity between different T2I blocks, it significantly reduces the additional learnable parameter number at a minor performance cost.

The collaboration between the play-and-plug framework and the Group Reuse Mechanism forms MERGE, which efficiently and flexibly unleashes the powerful zero-shot depth estimation potential of the pre-trained T2I model while retaining its inherent image generation capability.

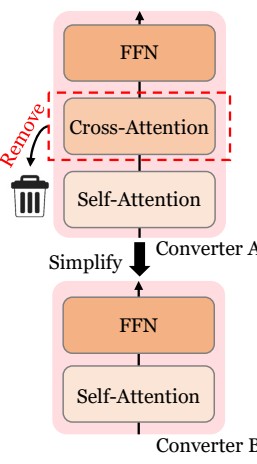

Converter simplified

Figure 5: The process of converter simplification, exemplified by PixArt [5].

### 4.3 Empirical Investigation

To smoothly transform the pre-trained T2I model into a generative depth estimation model through MERGE, the converter in MERGE is copied from the first T2I block of the respective group in the pre-trained T2I model, as illustrated by Converter A in Fig. 5. These converters are designed for T2I tasks and generally include components for multimodal feature interaction, such as Cross-Attention. However, existing diffusion-based generative depth estimation methods typically set the text prompt to empty. Intuitively, we suspect that the multimodal feature interaction design in the converter might be redundant for MERGE, as an empty text prompt contains no useful information. The experiments in Sec. 5.3 confirm this hypothesis, demonstrating that directly removing this multimodal interaction design, as illustrated by Converter B in Fig. 5, has almost no effect on the depth estimation performance of MERGE. This simplified converter reduces the 25% learnable parameter number in MERGE used to unleash depth estimation capability.

Similarly, in some of the latest MMDiT-based methods, like FLUX [19], a dual-stream multimodal interaction block can be simplified into a single-stream block to serve as a converter for MERGE. Using these simplified converters, MERGE can naturally leverage the rich visual priors of the pre-trained T2I model to demonstrate powerful zero-shot depth estimation capability while preserving its inherent image generation ability, which full-parameter fine-tuning methods cannot achieve.

## 5 Experiments

**Dataset and Evaluation metrics.** Following prior work [16, 20, 21], we train our model on the Hypersim [37] and Virtual KITTI [3] datasets, which correspond to synthetic indoor and outdoor datasets, respectively, comprising a total of 74k training samples. Subsequently, we evaluate its depth estimation performance on three real-world datasets: NYUv2 [40], ScanNet [6], and DIODE [43]. The quantitative evaluation metrics include absolute relative error (A.Rel) and the percentage of inlier pixels ($\delta1$) with thresholds of 1.25.

**Implementation Details.** We implement MERGE using PyTorch and employ PixArt-XL-2-512$\times$512 and FLUX.1-dev as our pre-trained text-to-image (T2I) models. The training is solely conducted on depth estimation data to build a unified image generation and depth estimation model. Following previous work [16], we double the input channel of the patchify layer, enabling it to handle image conditions for generative depth estimation. During the depth estimation inference process, this patchify layer can seamlessly replace the patchify layer used in the T2I process. Regarding the group reuse mechanism, we adopt a default strategy of evenly dividing groups to avoid model-specific designs. Training our MERGE takes 30K iterations using a batch size of 32. We use the Adam optimizer with learning rates of 1e-4 and 3e-4 for PixArt and FLUX, respectively. Additionally, we follow the defaults provided in [16] for depth data preprocessing settings. For hyperparameters not mentioned in the MERGE training process, we follow the fine-tuning settings provided by the official pre-trained T2I model. All of our experiments are implemented on 8 NVIDIA H20 GPUs.

### 5.1 Main Results

We present two versions of MERGE with different parameter scales using the pre-trained PixArt [5] and FLUX [19] models, referred to as MERGE-B and MERGE-L, respectively. Unless otherwise specified, the number of groups for MERGE-B and MERGE-L is set to 14 and 10, respectively.

**Compared with the state-of-the-art.** As shown in Tab. 1, MERGE-B achieves superior performance over JointNet [52] and UniCon [21] with only 110M additional learnable parameters. Distinguishing these methods that require running dual diffusion models in parallel, the play-and-plug framework of MERGE is more computationally efficient, as it only injects fewer converters in a sequential manner when needed. Notably, due to the feature interaction design between the parallel diffusion models, it registers new knowledge into the pre-trained T2I model, which affects its original image generation capability, whereas MERGE does not.

Compared to the latest work, OneDiffusion [20], which is driven by 100M data, our MERGE-L achieves superior performance, surpassing OneDiffusion by 0.9 A.Rel on NYUv2 and 1.1% $\delta1$ on DIODE, while requiring only approximately 12% additional parameters of the pre-trained T2I model. Remarkably, the learnable parameter of MERGE needs only about half the learnable parameters of OneDiffusion. By leveraging the powerful visual knowledge stored in the fixed pre-trained T2I

Table 1: Quantitative comparisons on zero-shot depth estimation. We compare our MERGE with works capable of both image generation and depth estimation on zero-shot depth estimation benchmarks. "-B" and "-L" refer to MERGE based on the pre-trained PixArt [5] and FLUX [19] models, respectively. "#Param." refers to the learnable parameter number, and "($\cdot$%)" in "#Param." represents the proportion of trainable parameters relative to the size of the original model. **Bold** numbers are the best.

| Method | Reference | Training Data | #Param. | NYUv2 | | ScanNet | | DIODE | |
|---|---|---|---|---|---|---|---|---|---|
| | | | | A.Rel $\downarrow$ | $\delta1 \uparrow$ | A.Rel $\downarrow$ | $\delta1 \uparrow$ | A.Rel $\downarrow$ | $\delta1 \uparrow$ |
| *These discriminative methods have* **depth estimation** *capability only.* | | | | | | | | | |
| DPT [35] | ICCV 21 | 1.2M | 123M(100%) | 9.8 | 90.3 | 8.2 | 93.4 | 18.2 | 75.8 |
| HDN [51] | NeurIPS 22 | 300K | 123M(100%) | 6.9 | 94.8 | 8.0 | 93.9 | 24.6 | 78.0 |
| DepthAnything [48] | CVPR 24 | 63.5M | 335M(100%) | 4.3 | 98.1 | 4.2 | 98.0 | 27.7 | 75.9 |
| DepthAnythingv2 [49] | NeurIPS 24 | 62.5M | 1.3B(100%) | 4.4 | 97.9 | - | - | 6.5 | 95.4 |
| *These generative methods have* **depth estimation** *capability only.* | | | | | | | | | |
| Marigold [16] | CVPR 24 | 74K | 889M(100%) | 5.5 | 96.4 | 6.4 | 95.1 | 30.8 | 77.3 |
| GeoWizard [8] | ECCV 24 | 280K | 889M(100%) | 5.2 | 96.6 | 6.1 | 95.3 | 29.7 | 79.2 |
| DepthFM [10] | AAAI 25 | 63K | 889M(100%) | 6.5 | 95.6 | - | - | 22.5 | 80.0 |
| Lotus [11] | ICLR 25 | 59K | 889M(100%) | 5.4 | 96.8 | 5.9 | 95.7 | 22.9 | 72.9 |
| *These generative methods support both* **image generation** *and* **depth estimation**. | | | | | | | | | |
| JointNet [52] | ICLR 24 | 65M | 889M(100%) | 13.7 | 81.9 | 14.7 | 79.5 | - | - |
| UniCon [21] | ICLR 25 | 16K | 125M(15%) | 7.9 | 93.9 | 9.2 | 91.9 | - | - |
| OneDiffusion [20] | CVPR 25 | 100M | 2.8B(100%) | 6.8 | 95.2 | - | - | **29.4** | 75.2 |
| MERGE-B (ours) | - | 74K | 110M(18%) | 7.5 | 94.2 | 9.9 | 89.8 | 32.5 | 74.9 |
| MERGE-L (ours) | - | 74K | 1.4B(12%) | **5.9** | **95.4** | **7.1** | **94.0** | 31.4 | **76.3** |

model, MERGE unifies image generation and depth estimation with less than one-thousandth of OneDiffusion's training data scale. Moreover, MERGE demonstrates overall superior depth estimation capability compared to OneDiffusion.

In addition, in the qualitative comparison, as shown in Fig. 6, MERGE demonstrates superior visual results, particularly in hollow and reflective regions, as highlighted by the black boxed areas.

**Compared with efficient low-rank fine-tuning methods.** We also present the comparison results between the efficient low-rank fine-tuning method and MERGE based on the same pre-trained T2I model. To ensure a fair comparison, we adjust the rank to match the number of learnable parameters in MERGE-B. The results shown in Tab. 2, compared to LoRA [13] and DoRA [28], MERGE-B performs better on all datasets. Unlike LoRA and DoRA, which fine-tune parameters at a finer-grained layer level, MERGE shows a structured converter at a higher level to enable depth estimation ability and proposes a group reuse mechanism to improve parameter utilization.

**Compared with full-parameter fine-tuning.** To further demonstrate the excellent zero-shot depth estimation capability of MERGE, Tab. 3 presents a comparison with the Marigold [16] paradigm, a generative depth estimation approach that full-parameter fine-tunes the pre-trained T2I diffusion model. We use the same pre-trained T2I model to present the PixArt version of Marigold (Marigold-P). As shown in Tab. 3, MERGE-B achieves results highly comparable to the full-parameter fine-tuning used by Marigold while requiring only 18% additional learnable parameters. Notably, if the converter is injected before each pre-trained T2I block, referred to as MERGE-B-28, it performs better on the NYUv2 dataset with only 37% learnable parameters of Marigold-P. More importantly, MERGE retains the image generation capability of the pre-trained T2I model thanks to the play-and-plug framework, which the Marigold paradigm can not achieve due to full-parameter fine-tuning disrupting the original feature distribution.

**Compared with traditional discriminative methods.** In terms of quantitative performance, state-of-the-art traditional discriminative methods (e.g., DepthAnything v2 [49]) still lead in monocular depth estimation, as shown in Tab. 1, particularly on challenging benchmarks like DIODE [43] that

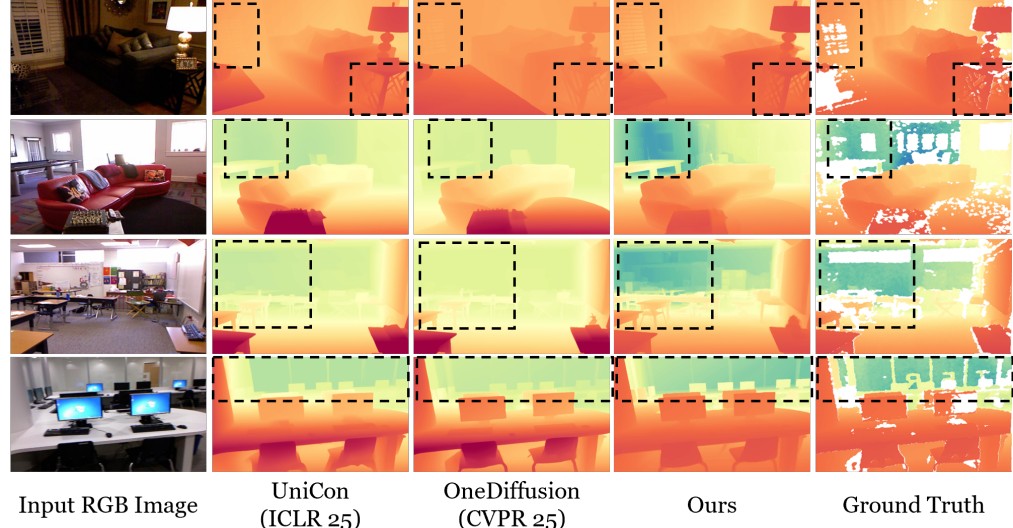

| Input RGB Image | UniCon
(ICLR 25) | OneDiffusion
(CVPR 25) | Ours | Ground Truth |
| --- | --- | --- | --- | --- |

Figure 6: Qualitative comparison between MERGE and other unified image generation and depth estimation methods on the NYUv2 benchmark. MERGE shows better depth estimation results, especially in detailed areas (such as free-form voids and reflect regions).

Table 2: Compared with low-rank fine-tuning methods based on the same text-to-image model [5].

| Method | Rank | #Param. | NYUv2 | | ScanNet | | DIODE | |
| --- | --- | --- | --- | --- | --- | --- | --- | --- |
| | | | A.Rel ↓ | $\delta1$ ↑ | A.Rel ↓ | $\delta1$ ↑ | A.Rel ↓ | $\delta1$ ↑ |
| LoRA [13] | 128 | 110M | 8.7 | 92.3 | 10.8 | 88.0 | 32.9 | 73.9 |
| DoRA [28] | 128 | 110M | 8.6 | 92.4 | 10.6 | 88.4 | 32.8 | 74.3 |
| MERGE-B | - | 110M | **7.5** | **94.2** | **9.9** | **89.8** | **32.5** | **74.9** |

Table 3: Compared with full-parameter fine-tuning. For a fair comparison, we adopt the same pre-trained T2I model to present the PixArt variant of Marigold [16], referred to as Marigold-P. "-28" indicates the number of groups is 28.

| Method | Support Tasks | | #Param. | NYUv2 | | ScanNet | | DIODE | |
| --- | --- | --- | --- | --- | --- | --- | --- | --- | --- |
| | Generation | Depth | | A.Rel ↓ | $\delta1$ ↑ | A.Rel ↓ | $\delta1$ ↑ | A.Rel ↓ | $\delta1$ ↑ |
| Marigold-P | ✗ | ✓ | 596M | 7.4 | 94.2 | 9.5 | 90.1 | **31.5** | **76.0** |
| MERGE-B | ✓ | ✓ | 110M | 7.5 | 94.2 | 9.9 | 89.8 | 32.5 | 74.9 |
| MERGE-B-28 | ✓ | ✓ | 224M | **7.0** | **94.7** | **9.2** | **91.1** | 31.8 | 75.5 |

include diverse outdoor scenes. However, it is important to note that the performance gap narrows significantly on indoor datasets, for example, our MERGE achieves highly competitive performance to DepthAnything v2 on the NYUv2 [40] benchmark (A.Rel: 4.4 *vs.* 5.9, $\delta1$: 97.9% *vs.* 95.4%).

## 5.2 Generalization to Normal Estimation Task

To further present the generality of our MERGE, we extend it to the zero-shot surface normal estimation. Following prior work [11], we employ the Hypersim [37] and Virtual KITTI [3] datasets as training data. Specifically, we filter out samples in Hypersim with invalid annotations and combine them with 20K Virtual KITTI training data, resulting in 59K training samples. All training hyperparameters are consistent with those used for the depth estimation task. We quantitatively evaluate the normal estimation performance of MERGE on NYUv2 [40], ScanNet [6], iBims-1 [18], and Sintel [2], reporting mean angular error ($m.$) as well as the percentage of pixels with an angular error below $11.25°$.

Table 4: Quantitative comparisons on normal estimation. "-B" and "-L" refer to MERGE based on the pre-trained PixArt [5] and FLUX [19] models, respectively. **Bold** numbers are the best.

| Method | Reference | Support Tasks | | NYUv2 | | ScanNet | | iBims-1 | | Sintel | |
|---|---|---|---|---|---|---|---|---|---|---|---|
| | | Gen. | Normal | $m. \downarrow$ | $11.25° \uparrow$ | $m. \downarrow$ | $11.25° \uparrow$ | $m. \downarrow$ | $11.25° \uparrow$ | $m. \downarrow$ | $11.25° \uparrow$ |
| Marigold [16] | CVPR 24 | ✗ | ✓ | 20.9 | 50.5 | 21.3 | 45.6 | 18.5 | 64.7 | - | - |
| GeoWizard [8] | ECCV 24 | ✗ | ✓ | 18.9 | 50.7 | 17.4 | 53.8 | 19.3 | 63.0 | 40.3 | 12.3 |
| StableNormal [50] | SIGGRAPH 24 | ✗ | ✓ | 18.6 | 53.5 | 17.1 | 57.4 | 18.2 | 65.0 | 36.7 | 14.1 |
| Lotus [11] | ICLR 25 | ✗ | ✓ | **16.5** | **59.4** | **15.1** | **63.9** | **17.2** | **66.2** | **33.6** | 21.0 |
| MERGE-B (ours) | - | ✓ | ✓ | 21.6 | 52.1 | 21.8 | 50.1 | 19.7 | 63.5 | 41.4 | 15.2 |
| MERGE-L (ours) | - | ✓ | ✓ | 20.1 | 54.3 | 18.3 | 60.2 | 18.2 | **66.2** | 36.9 | **21.3** |

Table 5: Ablation of the composition of converters. Here, GRE is disabled (i.e., a unique converter is inserted before each T2I block). SA/CA means the Self-Attention/Cross-Attention.

| Setting | SA | CA | FFN | FFN Scale | #Param. | A.Rel $\downarrow$ | $\delta1 \uparrow$ |
|---|---|---|---|---|---|---|---|
| A | ✓ | ✓ | ✓ | ×4 | 596M | 6.9 | 94.8 |
| B | ✓ | ✗ | ✓ | ×4 | 447M | 6.9 | 94.6 |
| C | ✗ | ✗ | ✓ | ×4 | 298M | 7.6 | 93.6 |
| D | ✓ | ✗ | ✗ | - | 149M | 7.4 | 94.2 |
| E | ✓ | ✗ | ✓ | ×1 | 224M | **7.0** | **94.7** |

As shown in Tab. 4, even compared with existing task-specific generative methods based on full-parameter fine-tuning, our MERGE-L achieves highly comparable or superior results across multiple benchmarks. More importantly, MERGE preserves the original image generation capability of the pre-trained text-to-image model, which others can not achieve due to full-parameter fine-tuning disrupting the original feature distribution.

### 5.3 Ablation Studies

Unless otherwise specified, we conduct ablation studies on PixArt [5] using the NYUv2 dataset [40], and the training hyperparameters remain consistent with those mentioned earlier.

**Ablation on the composition of the converter.** We first validate the hypothesis in Sec. 4.3 under the setting of not enabling GRE, where the multimodal interaction design, Cross-Attention, is redundant for MERGE's converters, as it fails to extract effective information from an empty text prompt. The results, shown in setting A and setting B of Tab. 5, demonstrate almost no performance difference between with and without Cross-Attention, while the latter reduces approximately $25\%$ the learnable parameter number. Meanwhile, we further explore the impact of other components in the converter, such as Self-Attention and the Feedforward Network (FFN). As demonstrated by the results of setting C and setting D, removing Self-Attention or FFN reduces the learnable parameters of the converter, and it significantly degrades the depth estimation performance.

It may be ascribed to these operations' sole focus on latent image features, and removing these components would significantly weaken the model's representational capacity. Additionally, an interesting finding is that reducing the expansion rate of the FFN from 4 to 1 in converters, as illustrated in setting E, causes almost no impact on performance. This significantly reduces the learnable parameter number of the converter by approximately $36\%$.

**Ablation on the Group Reuse Mechanism (GRE).** To demonstrate the effectiveness of the Group Reuse Mechanism, we divide two adjacent T2I blocks into a group, resulting in a total of 14 groups with no overlap between them. When the converter is fixedly inserted before the first T2I block of each group, i.e., without GRE, it causes a significant decrease in depth estimation performance (A.Rel from $7.0 \rightarrow 15.6$, $\delta1$ from $94.7 \rightarrow 78.8$), as shown in the first two rows of Tab. 6. However, when sharing a converter within the group, the same parameter number yields a significantly improved result (A.Rel from $15.6 \rightarrow 7.5$, $\delta1$ from $78.8 \rightarrow 94.2$).

Compared to inserting a converter before each pre-trained T2I block, GRE reduces the learnable parameter number by about half, with only a slight performance cost. Tab. 6 also presents the impact of different group divisions on depth estimation performance. The results show that as the number of divided groups increases, the depth estimation performance of MERGE gradually improves. It is reasonable since features between consecutive layers share similarities but still exhibit differences. With more groups, these differences are more effectively balanced.

Table 6: Ablation of Group Reuse Mechanism (GRE).

| Groups | GRE | #Param. | A.Rel ↓ | $\delta 1$ ↑ |
|--------|-----|---------|---------|--------------|
| 28 | ✗ | 224M | **7.0** | **94.7** |
| 14 | ✗ | 110M | 15.6 | 78.8 |
| 14 | ✓ | 110M | 7.5 | 94.2 |
| 7 | ✓ | 56M | 7.8 | 93.5 |
| 4 | ✓ | 32M | 9.3 | 91.0 |

Considering the trade-off between computational cost and performance, we ultimately adopt the setting of 14 groups as the default configuration for MERGE-B.

**Ablation on the number of converters.** We also demonstrate the impact of stacking varying numbers of converters on depth estimation performance. The results in Tab. 7 show that stacking multiple converters to create a larger one is ineffective and even has negative effects. We suspect this is due to the significant increase in model depth, making optimization extremely challenging.

Table 7: Ablation of converter number.

| Number | #Param. | A.Rel ↓ | $\delta 1$ ↑ |
|--------|---------|---------|--------------|
| 1 | 110M | **7.5** | **94.2** |
| 2 | 224M | 7.6 | 93.8 |
| 3 | 335M | 8.2 | 92.9 |

**Ablation on the initialization methods of the converter.** We further investigate the impact of different initialization methods for the converter on the depth estimation performance. The results in Tab. 8 show that initializing the converter with the pre-trained T2I block (the first T2I block of each group) consistently outperforms random initialization. This initialization strategy enables the converter to process the input features seamlessly.

Table 8: Ablation of initialization type.

| Init. Type | A.Rel ↓ | $\delta 1$ ↑ |
|------------|---------|--------------|
| Random | 7.9 | 93.5 |
| Pre-trained | **7.5** | **94.2** |

**Ablation on different text prompts.** We finally investigate the impact of different types of text prompts on the depth estimation performance of MERGE. Specifically, we conduct experiments using three types of prompts: the default empty prompt, a fixed "depth map" prompt, and dense captions generated by LLaVA-Lightning-MPT-7B [27]. As shown in Tab. 9, our results demonstrate that more specific captions provide a performance boost for our method. Considering the trade-off between the additional cost of obtaining dense captions and the resulting benefits, we utilize an empty prompt by default.

Table 9: Ablation of the text prompt.

| Prompt Type | A.Rel ↓ | $\delta 1$ ↑ |
|-------------|---------|--------------|
| Empty prompt | 7.5 | 94.2 |
| "Depth map" | 7.4 | 94.3 |
| Dense caption | **7.3** | **94.4** |

## 6  Conclusion

This paper presents MERGE, a method that starts from a fixed pre-trained text-to-image (T2I) model and unleashes its depth estimation capability while preserving its inherent image generation ability. Specifically, MERGE introduces a unified model without bells and whistles, which can seamlessly switch between original image generation and depth estimation modes through pluggable converters. Meanwhile, through empirical research, MERGE presents a simple and effective converter. Moreover, considering the similarity of output features across pre-trained T2I blocks, MERGE proposes a Group Reuse Mechanism to encourage parameter reuse, significantly reducing the additional parameter number. Extensive experiments demonstrate the effectiveness of MERGE. MERGE presents a new perspective for unifying multiple generative tasks besides the existing approaches that rely on massive data. Meanwhile, this play-and-plug strategy offers a cost-effective solution for extending the functionality of existing generative models.

**Limitation.** Despite the promising results achieved, our method still has limitations. For example, performing semantic segmentation using MERGE is highly challenging. One possible solution is to map semantic IDs to the RGB space [20]. However, the randomness of the denoising model makes it difficult to achieve a stable and accurate segmentation result by one-shot ID mapping from generated RGB results. In addition, integrating bit encoding methods, e.g., LDMSeg [42], is incompatible with the VAE of existing pre-trained text-to-image diffusion models. We consider this an important direction for future exploration.

**Acknowledgement.** This work was supported by the NSFC (62225603, 62441615, and 623B2038).

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
