# OpenReview forum: "More Than Generation: Unifying Generation and Depth Estimation via Text-to-Image Diffusion Models"
_NeurIPS.cc/2025/Conference — NeurIPS 2025 poster_

### Official Review · Reviewer_VXik · 2025-06-23

**Clarity:** 3
**Significance:** 2
**Originality:** 2
**Rating:** 4
**Confidence:** 4

**Summary:**

This paper introduces MERGE, a plug-and-play framework that transforms a fixed, pre-trained text-to-image (T2I) diffusion model into a unified model capable of both image generation and depth estimation. The authors achieve this unification by injecting lightweight, learnable converters before each transformer block (T2I block) in the diffusion model, allowing seamless switching between tasks without degrading the original generative quality. This paper also proposes the Group Reuse Mechanism (GRE), which reuses converters across similar T2I blocks to reduce the number of additional trainable parameters, improving parameter efficiency. MERGE demonstrates state-of-the-art performance on multiple real-world depth estimation benchmarks (e.g., NYUv2, ScanNet).

**Questions:**

1. The connection between the feature similarity shown in Figure 4 and the motivation for reusing converters is not clearly established, while the key motivation of this paper: how to unify depth estimation and image generation.
Besides, if the outputs of different T2I blocks are indeed highly similar, does this suggest redundancy in the DiT model’s internal representations? A deeper analysis would strengthen the justification for the Group Reuse Mechanism.

2. In addition, the proposed augmentation strategy appears to lack generalizability across architectures beyond PixArt, such as Stable Diffusion or FLUX, which raises concerns about the robustness of the approach.

**Ethical Concerns:**

["NO or VERY MINOR ethics concerns only"]

**Final Justification:**

I thank the authors for their efforts on the manuscript and rebuttal. Before the rebuttal, I raised concerns about generalization, unfair comparisons with baselines using Stable Diffusion, the motivation for reusing converters, and \etc. The authors have addressed these concerns well in their response. Therefore, I recommend this work as borderline accept.

**Limitations:**

yes.

**Paper Formatting Concerns:**

No obvious formatting concerns.

**Quality:**

3

**Strengths And Weaknesses:**

I thank the authors for their efforts in this work. Below are some comments.

**Strengths**

1. This manuscript is well-written and easy to follow.
2. The authors perform extensive ablation experiments (Section 5.2), systematically evaluating key design choices including converter composition, Group Reuse Mechanism (GRE), initialization strategies, and prompt settings. This validates the effectiveness of the proposed method.

**Weakness**

1. The comparison with baselines is not entirely fair, as the proposed method is based on PixArt, whereas several of the compared baselines, such as JointNet or UniCon, are built on different models like Stable Diffusion.
2. While the proposed converter introduces an incremental innovation by combining reused T2I model blocks with LoRA-like principles, the justification in Figure 4 appears model-specific and does not clearly generalize to architectures such as Stable Diffusion or FLUX. More discussion is included in # Question parts.
3. The idea of the proposed converter is incremental; it is essentially a combination of reusing T2I model blocks and LoRA.
4. While the paper claims to preserve the original image generation ability of the T2I model compared to the related work, it lacks quantitative or visual evaluations of image quality for the image generation tasks. This undermines a central claim of the work.
5. The ablation on the number of converters appears to violate scaling laws, yet no in-depth analysis is provided to explain this phenomenon. The paper also lacks a clear discussion on the trade-off between depth estimation and image generation capabilities. Furthermore, there is no ablation study exploring configurations with fewer than 110M learnable parameters.

---

> ### Author Rebuttal · Authors · 2025-07-30
>
> 1. **To weakness 1: “The comparison with baselines is not entirely fair, … such as JointNet or UniCon, are built on different models like Stable Diffusion.”**
>
>     **Reply:** Thanks. In the table below, we provide experiments on the Stable Diffusion version of MERGE (MERGE-SD). The experimental results demonstrate that MERGE-SD consistently outperforms JointNet and UniCon across multiple depth estimation benchmarks. We fully validate the advantages of our MERGE on multiple pre-trained Text-to-Image models, including Stable Diffusion, PixArt, and FLUX.
>
>     | Method | NYUv2-A.Rel ↓ | NYUv2-δ1 ↑ | Scannet-A.Rel ↓ | Scannet-δ1 ↑ |
>     | --- | --- | --- | --- | --- |
>     | JointNet | 13.7  | 81.9  | 14.7  | 79.5 |
>     | UniCon | 7.9  | 93.9 | 9.2 | 91.9 |
>     | MERGE-SD (Ours) | 6.1 | 95.6 | 7.6 | 93.2 |
> 2. **To weakness 2&3: “… converter introduces an incremental innovation by combining reused T2I model blocks with LoRA-like principles, the justification in Figure 4 appears model-specific and does not clearly generalize to architectures such as Stable Diffusion or FLUX. …”**
>
>     **Reply:** Thanks. Regarding your opinion that our converter design represents "incremental innovation," we would like to clarify that it fundamentally differs from methods like LoRA in motivation and architecture. **1) The core motivation is fundamentally different.** The motivation of LoRA stems from prior work [1], which revealed that pre-trained models possess an extremely low-dimensional parameter space. Fine-tuning within this space achieves the same effect across the entire parameter space. Essentially, LoRA provides an efficient implementation of full fine-tuning. Our intuition behind designing the converter is based on the fact that diffusion-based image generation and generative geometry estimation both belong to the latent diffusion process. The transition from image generation to geometry estimation may only lack a "task-oriented" converter, which guides the latent diffusion of image generation towards geometry estimation. **2) The architectural implementation has significant distinctions.** LoRA introduces low-rank matrices into existing layers to achieve efficient fine-tuning. In contrast, our approach injects a highly structured converter similar to Text-to-Image (T2I) blocks. The most direct advantage of our highly structured converter is that it enables converters to be initialized with the weights of pre-trained T2I models for parameter initialization, which LoRA does not support. The experimental results in the table below demonstrate that this initialization consistently outperforms random parameter initialization.
>
>     | Init.Type | NYUv2-A.Rel ↓ | NYUv2-δ1 ↑ |
>     | --- | --- | --- |
>     | Random | 7.9  | 93.5 |
>     | Pre-trained | 7.5  | 94.2 |
>
>     The motivation for our Group Reuse Mechanism (GRE) is inspired by observing the common characteristics of large Transformer models, such as Fig. 1 in [2], Fig. 3 in [3], and Fig. 4 in [4]. Specifically, adjacent layers often have a high degree of feature similarity. Fig. 4 in our main text uses PixArt as a visual example of this common phenomenon, rather than as comprehensive evidence.
>
> 3. **To weakness 4: “… lacks quantitative or visual evaluations of image quality for the image generation tasks. …”**
>
>     **Reply:** Good suggestion! The design of our method is to ensure the complete preservation of image generation capabilities. Specifically, all additionally introduced converters are skipped when performing the "text-to-image" task. It means that the inference process of image generation remains fully consistent with the original, unmodified, pretrained T2I model. Following your suggestion, we provide the quantitative evaluation results for image generation below. In addition, we present some images generated by our MERGE model in Fig. 1 of the main text. We sincerely thank your suggestion and commit to explicitly including the quantitative evaluation results of image generation in the final version of the paper to strengthen our core claims.
>
>     | Method | COCO FID-30K ↓ | GenEval ↑ |
>     | --- | --- | --- |
>     | SDXL | 6.63 | 0.55 |
>     | SD3-medium | 11.92 | 0.62 |
>     | JointNet | 14.15 | - |
>     | OneDiffusion | - |  0.65 |
>     | PixArt-α | 7.32 |  0.48 |
>     | MERGE-PixArt (Ours) | 7.32 |  0.48 |
>     | FLUX.1 dev | 10.15 |  0.67 |
>     | MERGE-FLUX (Ours) | 10.15 |  0.67 |
> 4. **To weakness 5: “… the number of converters appears to violate scaling laws, yet no in-depth analysis is provided to explain this phenomenon. … discussion on the trade-off between depth estimation and image generation capabilities. … no ablation study exploring configurations with fewer than 110M learnable parameters.”**
>
>     **Reply:** Thanks. Since converters are inserted into the pre-trained T2I model in a sequential manner, increasing the number of converters sharply increases the network depth. For example, when the number of converters is 3, the network depth expands 4x, which can lead to optimization challenges[5][6] and ultimately result in a decline in performance.
>
>     It is an excellent question regarding the "trade-off." Thanks to our plug-and-play design, our image generation capability remains constant and does not vary with changes in depth estimation performance.
>
>     We present the ablation study with fewer than 110M parameters in Table 5, as shown in the table below. In this table, we explore settings with 56M (7 groups) and 32M (4 groups) learnable parameters by adjusting the number of groups in the group reuse mechanism (GRE). The experimental results show that our method remains effective under a lower parameter budget.
>
>     | Groups  | GRE  | #Param. | NYUv2-A.Rel ↓ | NYUv2-δ1 ↑ |
>     | --- | --- | --- | --- | --- |
>     | 28  | × | 224M | 7.0  | 94.7 |
>     | 14 | × | 110M | 15.6  | 78.8 |
>     | 14 | ✔ | 110M | 7.5  | 94.2 |
>     | 7 | ✔ | 56M | 7.8  | 93.5 |
>     | 4 | ✔ | 32M |  9.3  | 91.0 |
> 5. **To question 1: “… feature similarity shown in Figure 4 and the motivation for reusing converters is not clearly established. … redundancy in the DiT model’s internal representations? …”**
>
>     **Reply:** Thanks. Fig. 4 of the main text reveals the high similarity between the output features of adjacent layers in the DiT model. Our idea is that if T2I block i and i+1 outputs are similar, the feature transformation for handling depth estimation tasks should also be similar. Therefore, sharing a converter may largely capture this common transformation.
>
>     Regarding the question of “redundancy”, we argue that it might highlight the widespread phenomenon of redundancy in the DiT model’s internal representations. We also note that a recent study[7] demonstrates that a significant portion of computation in global self-attention mechanisms is wasted on calculating near-zero attention scores between distant tokens, resulting in only a limited number of effectively updated features. Moreover, as Transformer models become deeper, feature representations in the deeper layers become overly similar[6], leading to a loss of diversity and expressive power, making additional layers computationally redundant. These findings indicate that the DiT model indeed contains redundancy.
>
> 6. **To question 2: “…lack generalizability across architectures beyond PixArt… Stable Diffusion or FLUX, …”**
>
>     **Reply:** Good suggestion! We would like to clarify that we discuss MERGE based on PixArt and FLUX in the paper. Additionally, we present the performance of MERGE based on Stable Diffusion in the table below. Our three variants achieve outstanding performance on multiple depth estimation benchmarks. These results directly demonstrate the strong generalizability of our method, which is not limited to the PixArt architecture alone.
>
>     | Method | NYUv2-A.Rel ↓ | NYUv2-δ1 ↑ | Scannet-A.Rel ↓ | Scannet-δ1 ↑ | DIODE-A.Rel ↓ | DIODE-δ1 ↑ |
>     | --- | --- | --- | --- | --- | --- | --- |
>     | JointNet | 13.7  | 81.9  | 14.7  | 79.5 | - | - |
>     | UniCon | 7.9  | 93.9 | 9.2 | 91.9 | - | - |
>     | OneDiffusion | 6.8  | 95.2 | - | - | 29.4  | 75.2 |
>     | MERGE-PixArt (Ours) | 7.5 | 94.2 | 9.9 | 89.8 | 32.5 | 74.9 |
>     | MERGE-Stable-Diffusion (Ours) | 6.1 | 95.6 | 7.6 | 93.2 | 30.8 | 93.4 |
>     | MERGE-Flux (Ours) | 5.9  | 95.4  | 7.1 | 94.0 | 31.4 | 76.3 |
>
> [1] Intrinsic Dimensionality Explains the Effectiveness of Language Model Fine-Tuning, ACL 2021.
>
> [2] Do Vision Transformers See Like Convolutional Neural Networks?, NeurIPS 2021.
>
> [3] Skip-Attention: Improving Vision Transformers by Paying Less Attention, ICLR 2024.
>
> [4] Layerlink: Bridging remote sensing object detection and large vision models with efficient fine-tuning, PR 2025.
>
> [5] Going Deeper with Image Transformers, ICCV 2021.
>
> [6] Swin Transformer V2: Scaling Up Capacity and Resolution, CVPR 2022.
>
> [7] Swin DiT: Diffusion Transformer using Pseudo Shifted Windows, Arxiv 2025.

---

> > ### Comment · Reviewer_VXik · 2025-08-04
> >
> > Thank you for your efforts in the response. My concerns have been well addressed, and I have raised my rating from 2 to 4.

---

> > > ### Author Response · Authors · 2025-08-04
> > >
> > > We sincerely thank you for the feedback and support. Your comments are valuable to us in improving the quality of this work. We will incorporate your suggestions and clarify key points in the revision. Many thanks again!

---

### Official Review · Reviewer_FL3u · 2025-07-03

**Clarity:** 3
**Significance:** 3
**Originality:** 3
**Rating:** 4
**Confidence:** 3

**Summary:**

This paper proposes a unified model that can generate images and estimate depth. Starting from a pretrained text-to-image model, this work introduces a play-and-plug framework that can switch between different tasks. It also proposes a group reuse mechanism for better utilization of parameters. Experimental results show the effectiveness of proposed methods in both tasks.

**Questions:**

Please refer to the weaknesses.

**Ethical Concerns:**

["NO or VERY MINOR ethics concerns only"]

**Final Justification:**

I have carefully written my reviews and actively discussed with the authors in multiple rounds. I have also read the comments from other reviewers.

I gave a score of *borderline accept* because the paper proposes an interesting approach for depth estimation by introducing converters into diffusion models. This paper is well written.

I did not give a higher score because I believe this work is better categorized as a single-task method, similar to Lotus, rather than a unified model as claimed. When used for image generation, the proposed method degrades to the original diffusion model without any improvement.  With converters, the proposed method can only support depth estimation.  In contrast, unified models such as OneDiffusion support a broader range of tasks, including segmentation, pose estimation, ID customization, and so on.  Compared to unified models like OneDiffusion, this work supports much fewer tasks.  Compared to single-task methods like Lotus, the  performance on depth estimation is lower. Considering these points, I will maintain my score and not raise it to a 5.

**Limitations:**

This paper discusses the limitations.

**Paper Formatting Concerns:**

N/A.

**Quality:**

3

**Strengths And Weaknesses:**

**Strengths:**
1. The proposed MERGE method adopts a few converters to switch between tasks, which is simple and efficient.
2. This work conducts extensive experiments to show the effectiveness of the proposed method in both image generation and depth estimation tasks.
3. This paper is well-organized and easy to follow.

**Weaknesses:**
1.   This paper highlights that the proposed method uses less training data compared to prior studies such as OneDiffusion. However, it is not discussed whether the training data is a subset of that used in previous studies or not. For example, OneDiffusion uses 40K images from Hypersim dataset,  without using Virtual KITTI used in this work. Therefore, it is not clear whether the performance improvement comes from different training sets or the proposed method.

2.  This work shows the effectiveness of MERGE over prior studies in Table 1 and Figure 6. However, further analysis is preferred. For example, what kinds of scenarios is the proposed method particularly effective in?

3. The paper compares depth estimation with diffusion-based methods like Lotus (Table 1), but lacks the comparison to traditional methods like Depth Anything[1] and Depth Anything V2 [2]. This may mislead the readers on the results of the proposed methods.

[1] Depth Anything: Unleashing the Power of Large-Scale Unlabeled Data, CVPR 2024.
[2] Depth Anything V2. A More Capable Foundation Model for Monocular Depth Estimation, NeurIPS 2024.

---

> ### Author Rebuttal · Authors · 2025-07-30
>
> 1. **To weakness 1: “… OneDiffusion uses 40K images from Hypersim dataset, without using Virtual KITTI used in this work. … the performance improvement comes from different training sets or the proposed method.”**
>
>     **Reply:** Good question! To address the concern that the performance improvement might stem from different training sets, we train a MERGE-L model using only Hypersim as training data, without Virtual KITTI. The experimental results are shown in the table below. Compared to OneDiffusion, the MERGE-L model trained solely on Hypersim still achieves comparable or even better results on the depth estimation benchmarks. This result demonstrates that the performance gain is primarily attributable to our proposed MERGE method rather than a specific data combination. In addition, we would like to mention that the depth map training data for OneDiffusion includes the Hypersim dataset and approximately 500K high-quality data, while we use 74K depth map training data.
>
>     | Method | Depth Map Training Data | NYUv2-A.Rel ↓ | NYUv2-δ1 ↑ |  DIODE-A.Rel ↓ |  DIODE-δ1 ↑ |
>     | --- | --- | --- | --- | --- | --- |
>     | OneDiffusion | Hypersim+500K depth data | 6.8  | 95.2 | 29.4 | 75.2 |
>     | MERGE-L (Ours) | Hypersim | 6.2 | 95.2 | 31.0 | 76.5 |
> 2. **To weakness 2: “… what kinds of scenarios is the proposed method particularly effective in?”**
>
>     **Reply:** Good question! Our current experimental setup is an out-of-distribution evaluation, where training is performed on synthetic data and evaluation is conducted on real-world datasets. Experimental results indicate that, compared to other unified models, our method demonstrates superior performance in zero-shot scenarios, especially in indoor scenarios. Additionally, Appendix B provides qualitative comparison results under challenging objects (Reflective Objects, Free-form Voids, and Filiform Objects). These results show that MERGE exhibits unique advantages in these challenging objects. We will highlight these findings in the main text of the final manuscript.
>
> 3. **To weakness 3: “… lacks the comparison to traditional methods like Depth Anything[1] and Depth Anything V2 [2]. This may mislead the readers on the results of the proposed methods.”**
>
>     **Reply:** Good suggestion! To provide readers with a more complete overview of the current state of development in the depth estimation field, we commit to including the results of traditional methods like the Depth Anything series[1][2] in Table 1 in the final version.
>
>
> [1] Depth Anything: Unleashing the Power of Large-Scale Unlabeled Data, CVPR 2024.
>
> [2] Depth Anything V2. A More Capable Foundation Model for Monocular Depth Estimation, NeurIPS 2024.

---

> > ### Comment · Reviewer_FL3u · 2025-08-02
> >
> > Thanks to the authors for their rebuttal, which addresses most of my concerns. Regarding Weakness 3, could the authors provide a more detailed comparison between traditional depth estimation models and the proposed generative method? How do they perform under different scenarios and datasets？ What is the benefit of using diffusion-based models for depth estimation? Do diffusion-model priors bring some new knowledge for depth estimation, which traditional discriminative approaches may not have effectively learned?

---

> ### Author Response · Authors · 2025-08-03
>
> This is a highly valuable question that allows us to clarify the differences and advantages of our MERGE, compared to traditional depth estimation discriminative methods.
>
> **1. Question: How do they perform under different scenarios and datasets?**
>
> **Reply:** In terms of quantitative performance, state-of-the-art traditional discriminative methods (e.g., DepthAnything v2) still lead in monocular depth estimation,  as shown in the table below, particularly on challenging benchmarks like DIODE that include diverse outdoor scenes. However, it is important to note that the performance gap narrows significantly on indoor datasets, for example, our MERGE achieves highly competitive performance to DepthAnything v2 on the NYUv2 benchmark (A.Rel: 4.4 vs. 5.9, δ1: 97.9 vs. 95.4).
>
> **2. Question: What is the benefit of using diffusion-based models for depth estimation?**
>
> **Reply:** The benefits are primarily twofold: **1) Efficient training enabled by strong priors.** Generative depth estimation methods demonstrate impressive data efficiency by leveraging the powerful priors from pre-trained T2I models. For example, our MERGE achieves its competitive performance on NYUv2 while using only about 1% of the depth training data required by DepthAnything v2. **2) The same paradigm enables a unified framework.** The emergence of generative depth estimation methods[1][2] makes it possible to unify the previously distinct tasks of geometric estimation and image generation. Building on this idea, we explore a novel unified paradigm for image generation and geometric estimation within a simple framework, which is not achievable with traditional discriminative depth estimation methods.
>
> **3. Question: Do diffusion-model priors bring some new knowledge for depth estimation, which traditional discriminative approaches may not have effectively learned?**
>
> **Reply:** The most important “new knowledge” brought by the diffusion model prior is its deep semantic and structural understanding of the visual world. These models can generate realistic and previously unseen content, which indicates that they have internalized common sense about objects, scenes, and physical laws. This “knowledge” may help the model produce physically plausible and reasonable depth structures for depth estimation tasks, especially when dealing with challenging materials such as textureless or blurry regions and reflective or transparent surfaces[3]. This profound world understanding, demonstrated by the model’s generative capabilities, is something traditional discriminative methods may not be able to learn effectively.
>
> [1] Repurposing Diffusion-Based Image Generators for Monocular Depth Estimation, CVPR 2024.
>
> [2] GeoWizard: Unleashing the Diffusion Priors for 3D Geometry Estimation from a Single Image, ECCV 2024.
>
> [3] Lotus: Diffusion-based Visual Foundation Model for High-quality Dense Prediction, ICLR 2025.
>
> | Method | Reference | Pre-trained Data | Depth Training Data | Image Generation | NYUv2-A.Rel ↓（Indoor） | NYUv2-δ1 ↑（Indoor） | Scannet-A.Rel ↓（Indoor） | Scannet-δ1 ↑（Indoor） | DIODE-A.Rel ↓（Indoor-Outdoor） | DIODE-δ1 ↑（Indoor-Outdoor） |
> | --- | --- | --- | --- | --- | --- | --- | --- | --- | --- | --- |
> | **Discriminative** |  |  |  |  |  |  |  |  |  |  |
> | MiDaS | TPAMI 20 | 1M | 2M | × | 11.1  | 88.5 | 12.1  | 84.6 | 33.2  | 71.5 |
> | DPT | ICCV 21 | 14M | 1.2M | × | 9.8  | 90.3 | 8.2  | 93.4 | 18.2  | 75.8 |
> | HDN | NeurIPS 22 | 14M | 300K | × | 6.9  | 94.8 | 8.0  | 93.9 | 24.6  | 78.0 |
> | DepthAnything | CVPR 24 | 142M | 63.5M | × | 4.3  | 98.1 | 4.2  | 98.0 | 27.7  | 75.9 |
> | DepthAnythingv2 | NeurIPS 24 | 142M | 62.5M | × | 4.4  | 97.9 | - | - |  6.5  | 95.4 |
> | **Generative** |  |  |  |  |  |  |  |  |  |  |
> | Marigold | CVPR 24 | 2.3B | 74K | × | 5.5  | 96.4  | 6.4  | 95.1  | 30.8  | 77.3 |
> | GeoWizard | ECCV 24 | 2.3B | 280K | × | 5.2  | 96.6  | 6.1  | 95.3  | 29.7  | 79.2 |
> | DepthFM | AAAI 25 | 2.3B | 63K | × | 6.5  | 95.6 | - | -  | 22.5  | 80.0 |
> | Lotus | ICLR 25 | 2.3B | 59K | × | 5.4  | 96.8  | 5.9  | 95.7  | 22.9  | 72.9 |
> | **Unifiled** |  |  |  |  |  |  |  |  |  |  |
> | JointNet | ICLR 24 | 2.3B | 65M | ✔ | 13.7 | 81.9 | 14.7 | 79.5 | - | - |
> | OneDiffusion | CVPR 25 | 100M | 540K | ✔ | 6.8 | 95.2 | - | - | 29.4 | 75.2 |
> | MERGE-PixArt (Ours) | - | 25M | 74K | ✔ | 7.5 | 94.2 | 9.9 | 89.8 | 32.5 | 74.9 |
> | MERGE-Flux (Ours) | - | Not Released | 74K | ✔ | 5.9 | 95.4 | 7.1 | 94.0 | 31.4 | 76.3 |

---

> ### Comment · Reviewer_FL3u · 2025-08-05
>
> Thanks to the authors for the response. I have also read the comments from other reviewers. While the proposed method is based on diffusion models, I believe comparisons with discriminative approaches like DepthAnythingv2 are necessary to not only understand its relative performance but also to show the benefit of using diffusion models. It is recommended to include these discussions in the final paper.
>
> While I acknowledge the contribution of this work, categorizing it as a unified model may be overstated. For example, prior unified models, such as OneDiffusion support multiple tasks with a single model, including segmentation, pose estimation, ID customization, and so on. By contrast, this work retains the original diffusion model's image generation ability without further improvement.  After adding the converter, the proposed method can only support one task: depth estimation. Considering this, I prefer to categorize this work as a generative method, similar to Lotus. However, the proposed method underperforms prior generative approaches like Lotus. Considering these points, I will maintain my score as borderline accept, without raising it to a 5.

---

> ### Author Response · Authors · 2025-08-05
>
> We sincerely thank your careful review of our work. We will follow your suggestion and include a comparison and discussion with discriminative approaches such as DepthAnythingV2 in the final paper.
>
> Regarding your comment about “... may be overstated,” we would like to clarify that our approach is not limited to the depth estimation task. We have also extended it to normal estimation in the supplementary material. Furthermore, our method can be easily transferred to other tasks such as semantic segmentation and style transfer. We will continue to explore these tasks using our method in the future.
>
> Regarding Lotus, we would like to clarify that Lotus focuses on fully leveraging the priors of pre-trained T2I models to achieve better depth estimation. In contrast, our method explores how to extend the capabilities of generative models without compromising their image generation abilities. We hope our approach can provide more insights to the community.
>
> Once again, thank you for your time and valuable suggestions during the review process.

---

### Official Review · Reviewer_Xt7v · 2025-07-03

**Clarity:** 3
**Significance:** 2
**Originality:** 3
**Rating:** 4
**Confidence:** 3

**Summary:**

This paper introduces MERGE, an unified diffusion model designed for both image generation and depth estimation. MERGE leverages a fixed, pre-trained text-to-image diffusion model and enhances it with plug-and-play converters. Group Reuse Mechanism encourages parameter reuse and improves efficiency. The model demonstrates superior depth estimation performance while uniquely preserving the original image generation capabilities of the pre-trained T2I model.

**Questions:**

- Please refer to the weaknesses section.
- Would it be feasible to extend this framework to enable co-generation of an image and its corresponding depth map simultaneously from a text prompt?

**Ethical Concerns:**

["NO or VERY MINOR ethics concerns only"]

**Final Justification:**

The authors’ rebuttal has alleviated some of my concerns. The arguments regarding OOD performance help to strengthen the paper, and I am convinced that the proposed method demonstrates strong performance under the same problem setup.

I am not fully convinced that the problem setup—“depth inference while preserving image generation priors”—is a necessary direction for the field. As Reviewer RUJQ raised, the authors have not provided a convincing reason why it is essential to preserve image generation priors for depth generation. If specialized methods that do not retain the generative prior can achieve better performance, it would be more natural to use those methods. The actual advantage of performing both image generation and depth inference within a single model is not clearly presented in the paper.

(Update) Thanks to the authors’ additional comment, I have come to agree with the necessity of this problem setup to some extent. Considering the scalability of image generation models, I also agree that this could be a step toward more promising research directions in the future. While I would not raise my score above borderline accept, I am leaning toward acceptance and will maintain my current rating.

**Limitations:**

Yes

**Paper Formatting Concerns:**

I could not find any major formatting issues.

**Quality:**

2

**Strengths And Weaknesses:**

Strengths

- Enabling powerful depth estimation without irreversibly degrading the inherent image generation capability of the pre-trained DiT is impressive.
- The proposed Group Reuse Mechanism is a well-reasoned design choice that improves parameter utilization and efficiency, which outperforms while maintaining similar parameter counts.
- It demonstrates superior performance on multiple depth estimation benchmarks, especially when considering its significantly lower training data and additional parameter requirements compared to other unified models.
- The paper is written clearly and is easy to follow.

Weaknesses
- The paper would benefit from a more explicit and in-depth discussion on the primary motivation for preserving the pre-trained T2I model's image generation capability. While the paper mentions catastrophic degradation, the advantages of retaining the original generative ability when only depth estimation is needed are not fully highlighted. Specifically, the lower performance compared to Marigold, which is a full fine-tuning method without generation preservation, in Table 1 raises questions about the trade-off. Clarifying why a model capable of both image generation and depth estimation is important would strengthen this argument.

- Following on from the point above, if a key benefit of preserving the T2I prior is enhanced generalization capability, demonstrating its robustness on OOD data would further validate the importance of maintaining the generative prior.

- The comparison in Table 3 is relevant to the discussion on motivating T2I prior preservation. Providing more details on this would strengthen the paper's rationale.

---

> ### Author Rebuttal · Authors · 2025-07-30
>
> 1. **To weakness 1: “… motivation for preserving the pre-trained T2I model's image generation capability. … the lower performance compared to Marigold, … raises questions about the trade-off. Clarifying why a model capable of both image generation and depth estimation … .”**
>
>     **Reply:** Thanks. Recent works, such as Show-o[1], UniCon[2], and OneDiffusion[3], reveal a promising research direction that merges image generation capabilities with understanding or perception capabilities in a single model, which are originally independent. Thus, our method aims to explore a new paradigm that merges the previously independent image generation and geometric perception (e.g., depth estimation) capabilities into a unified model, rather than pushing the performance border of a single task. So, we would like to clarify that a fair and meaningful comparison should be made with unified-based methods like UniCon (ICLR 25) and OneDiffusion (CVPR 25) rather than with specialized models (e.g., Marigold). Notably, even compared to Marigold, our method shows only a minor performance gap (e.g., only 0.4 A.Rel gap on the NYUv2 benchmark). More importantly, our method preserves the original image generation capability of the pre-trained model, which Marigold does not. We believe that trading a minor difference in single-task performance to preserve the model's original image generation capability is a valuable trade-off for developing unified models for image generation and perception in the community.
>
> 2. **To weakness 2: “Following on from the point above, if a key benefit of preserving the T2I prior is enhanced generalization capability, demonstrating its robustness on OOD data would further validate the importance of maintaining the generative prior.”**
>
>     **Reply:** Good suggestion! We apologize for not clearly explaining the experiment setting. Actually, our current training on synthetic datasets and evaluation on three real-world datasets (NYUv2, ScanNet, and DIODE) aligns with the out-of-distribution (OOD) test setting. Our MERGE outperforms other unified models, such as JointNet (ICLR 24), UniCon (ICLR 25), and OneDiffusion (CVPR 25), on these real-world benchmarks, proving that preserving and leveraging the robust visual priors from Text-to-Image models is crucial for enhancing a model's zero-shot generalization capabilities. Thanks for your suggestion. We will explicitly clarify our experimental setup in the final version.
>
> 3. **To weakness 3: “The comparison in Table 3 is relevant to the discussion on motivating T2I prior preservation. Providing more details on this would strengthen the paper's rationale.**”
>
>     **Reply:** Thanks. Previous work [4] suggests that the catastrophic forgetting during the fine-tuning of Text-to-Image (T2I) models often leads to vague predictions in highly detailed areas. Thus, a naive idea is that preserving the T2I prior might alleviate this issue. So, we provide a direct and fair comparison between our method and Marigold under an identical T2I model. The results show that our MERGE not only achieves highly competitive and even better performance compared to the Marigold-P with fewer learnable parameters but also preserves the original generation capability, an advantage not supported by the full fine-tuning approach.
>
> 4. **To question 2: “Would it be feasible to extend this framework to enable co-generation of an image and its corresponding depth map simultaneously from a text prompt?”**
>
>     **Reply:** This is a very insightful question, and the answer is yes! In Fig. 1 of the main text, we present examples of generating an image and its corresponding depth estimation annotation using a two-stage method (text-to-image -> image-to-depth). A possible one-stage solution to simultaneously generate the image and its corresponding depth map involves branching the features for image generation and depth estimation. After passing through a shared Text-to-Image block, one branch converts the features for the depth estimation task via our converters, while the other branch skips the converters to complete the original image denoising process. This allows for the simultaneous output of both the image and its corresponding depth map. Follow your suggestion, we will refine this solution in our future work.
>
>
> [1] Show-o: One Single Transformer to Unify Multimodal Understanding and Generation, ICLR 2025.
>
> [2] A Simple Approach to Unifying Diffusion-based Conditional Generation, ICLR 2025.
>
> [3] One Diffusion to Generate Them All, CVPR 2025.
>
> [4] Lotus: Diffusion-based Visual Foundation Model for High-quality Dense Prediction, ICLR 2025.

---

> > ### Comment · Reviewer_Xt7v · 2025-08-05
> >
> > Thank the authors for the rebuttal. It has alleviated some of my concerns. The additional claims regarding OOD performance help to strengthen the paper, and I am convinced that the proposed method demonstrates strong performance under the same problem setup.
> >
> > However, I am still not fully convinced that the problem setup—“depth inference while preserving image generation priors”—is a necessary direction for the field. As Reviewer RUJQ raised, the authors have not provided a convincing reason why it is essential to preserve image generation priors for depth generation. If specialized methods that do not retain the generative prior can achieve better performance, it would be more natural to use those methods. The actual advantage of performing both image generation and depth inference within a single model is not clearly presented in the paper. It would be much appreciated if the authors could more clarify this aspect.

---

> > > ### Author Response · Authors · 2025-08-06
> > >
> > > Thank you for your feedback, which gives us an opportunity to further clarify the motivation behind our work.
> > >
> > > The growing number of specialized models has led to challenges in managing these systems efficiently and optimizing computational resources. A more scalable solution is a single, unified model capable of handling diverse tasks, simplifying both development and deployment. However, image generation and geometric estimation are typically regarded as separate tasks because of their distinct inherent paradigms. Recent studies, including UniCon (ICLR 2025), OneDiffusion (CVPR 2025), and UniVG (ICCV 2025), indicate that unifying these tasks may be a promising research direction, but they often introduce significant complexity through multi-stage training or architectural modifications. Thus, we aim to explore a new simple paradigm that extends the geometric estimation capability (e.g., depth estimation) of pre-trained T2I models without compromising their image generation capability. As the priors of T2I models continue to strengthen, we believe that our approach will soon catch up with, and even surpass, the performance of specialized methods.
> > >
> > > Given the reasons above, comparing our method with specialized methods may not be reasonable. A meaningful comparison should be made with methods that also unify image generation and depth estimation tasks, like OneDiffusion. Meanwhile, experiments presented in the supplementary materials show that our method achieves overall better results than specialized methods on the normal estimation task.
> > >
> > > Considering the early exploration of extending the geometric estimation capabilities of pre-trained T2I models without compromising their image generation abilities, we believe our work is valuable to researchers in this field and hope it provides new insights for future research.
> > >
> > > [1] A Simple Approach to Unifying Diffusion-based Conditional Generation, ICLR 2025.
> > >
> > > [2] One Diffusion to Generate Them All, CVPR 2025.
> > >
> > > [3] UniVG: A Generalist Diffusion Model for Unified Image Generation and Editing, ICCV 2025

---

> > > > ### Comment · Reviewer_Xt7v · 2025-08-06
> > > >
> > > > Thank you for the additional comment, which helped me better understand the motivation. I will take this into account in the final rating!

---

> > > > > ### Author Response · Authors · 2025-08-06
> > > > >
> > > > > We sincerely thank you for your time and constructive feedback. Your comments are valuable for us to improve the quality of our paper. We will incorporate your suggestions and clarify key points in the final paper. Thank you once again!

---

### Official Review · Reviewer_RUJQ · 2025-07-08

**Clarity:** 3
**Significance:** 1
**Originality:** 2
**Rating:** 4
**Confidence:** 3

**Summary:**

Authors present a unified framework for image generation and depth estimation with a fixed T2I diffusion model. Specifically, they introduce an identical, learnable T2I block as a converter before each transformer layer of the pre-trained T2I diffusion transformer. The play-and-plug design enables seamless switching between the original T2I model and depth estimation model by skipping or running these converters. Moreover,considering the similarity of output features across pre-trained T2I blocks, MERGE proposes a Group Reuse Mechanism (GRE) to encourage parameter reuse that significantly reduces the number of parameters. Extensive experiments demonstrate the effectiveness of MERGE for the task of depth estimation compared to existing approaches that rely on  massive data.

**Questions:**

- I would suggest to extend the work to at least one more generation task other than style transfer like tasks.

**Ethical Concerns:**

["NO or VERY MINOR ethics concerns only"]

**Final Justification:**

Comment:
Authors added another normal estimation analysis in the rebuttal and demonstrated better performance than the compared methods, which somewhat mitigates my concerns of the generalization ability to other image generation tasks. However, in my initial review, I suggest to see some experiment results other than style transfer like tasks. As the normal estimation is very similar to depth information, using stuctural information from original image to convert the pixel values to another format, I am not completely convinced at this point. The proposal and rebuttal are 100% clear delivering the message that the framework is capable of maintaining good image generation abilities across similar tasks, but the rebuttal seems avoid answering the core question raised in the review. One thing I may overlooked in the inital review is the evaluation of out-of-distribution (OOD). I think this part the proposal did a good job and should add values to the final score. So I lift the rating to boarderline accept.

**Limitations:**

yes

**Quality:**

2

**Strengths And Weaknesses:**

Strengths:
- Plug-and-play design offers flexibility of the network capability and such design can theoretically extend to other image modality.
- Thorough ablation studies of the the converter to show the effectiveness of the proposed design details.

Weakness:
- The research topic is worth discussing. As diffusion models by default are good at image style transfer, it is not surprising to levarage the generation power to do the depth estimation as depth maps can be treated as another style of the image with same structures. As the motivation of doing depth estimation with diffusion network does not bring much novelty, the next question is whether the proposal brings a better depth estimation solution. However, from the Table 1 comparison we can see the depth estimation expert models (Marigold, GeoWizard and etc.) outperforms the proposed method with less training parameters and close quantity of training data. So for real world applications, users can simply load two models to have both functionalities other than switching on and off the converter. Even though the study in Table 2 shows that the proposed method is better the Lora/Dora fine-tuning methods, indicating an superior design of GRE, the value of the proposal in real world senarios is not clear unless the work shows the possibilities to extend the work to other image generation tasks, like OneDiffusion.
- Minor typo. Plug-and-play and play-and-plug are both used in the text.

---

> ### Author Rebuttal · Authors · 2025-07-30
>
> 1. **To weakness 1: “… not surprising to levarage the generation power to do the depth estimation … comparison we can see the depth estimation expert models (Marigold, GeoWizard and etc.) … . So for real world applications …”**
>
>     **Reply**: Good question! In general, image generation and geometric estimation (e.g., depth estimation) are considered two totally different tasks. The emergence of recent works in the generative field, such as Show-o[1], UniCon[2], and OneDiffusion[3], indicates that the unified of previously separate tasks, including understanding, generation, and perception, is a promising research direction. So, our method aims to explore **a new paradigm for constructing a unified model for image generation and geometric perception in a simple framework, instead of simply pushing the performance border of a single task.**
>
>     Although methods like Marigold and GeoWizard achieve strong performance through full fine-tuning, their image generation capability is inevitably compromised. Thus, a relatively fair comparison would be with UniCon[2] (ICLR 25) and OneDiffusion[3] (CVPR 25), which also possess image generation and geometric perception capabilities. As shown in the table, even when compared to Marigold, our A.Rel metric shows a difference of less than 1 across three depth estimation benchmarks. Moreover, in the evaluation of normal estimation, our MERGE achieves an overall better performance. More importantly, our MERGE retains the capability for image generation, which Marigold does not.
>
>     Considering the early exploration of merging geometric perception capabilities while preserving the original image generation capabilities of pre-trained models, we believe our work is valuable to researchers in this field.
>
>     | Method\Depth | NYUv2-A.Rel ↓ | NYUv2-δ1 ↑ | Scannet-A.Rel ↓ | Scannet-δ1 ↑ | DIODE-A.Rel ↓ | DIODE-δ1 ↑ |
>     | --- | --- | --- | --- | --- | --- | --- |
>     | Marigold | 5.5  | 96.4  | 6.4  | 95.1  | 30.8  | 77.3 |
>     | GeoWizard  | 5.2  | 96.6  | 6.1  | 95.3 | 29.7 | 79.2 |
>     | MERGE (Ours) | 5.9 | 95.4 | 7.1 | 94.0 | 31.4 | 76.3 |
>
>     | Method\Normal | NYUv2-m.↓ | NYUv2-11.25◦↑ | Scannet-m.↓ | Scannet-11.25◦↑ |  iBims-1-m.↓ |  iBims-1-11.25◦↑ |
>     | --- | --- | --- | --- | --- | --- | --- |
>     | Marigold | 20.9  | 50.5  | 21.3  | 45.6  | 18.5  | 64.7 |
>     | GeoWizard  | **18.9** | 50.7 | **17.4** | 53.8 | 19.3  | 63.0 |
>     | MERGE (Ours) | 20.1 | **54.3**  | 18.3  | **60.2**  | **18.2**  | **66.2** |
> 2. **To weakness 1: “… the value of the proposal in real world senarios is not clear unless the work shows the possibilities to extend the work to other image generation tasks … .”**
>
>     **Reply**: Good suggestion! We would like to clarify that our current experiments align with the evaluation of out-of-distribution (OOD), which involves training on synthetic datasets and testing on real-world datasets. The experiment results represent impressive performance, demonstrating the potential of our method in the real world. Additionally, following your suggestion, we extend our method to the normal generation task. Please refer to Appendix A.2 for specific details. Our method achieves highly competitive or even better performance than current state-of-the-art full fine-tuning models. Critically, these models lose their image generation capabilities after fine-tuning, whereas our MERGE method preserves the original text-to-image generation ability.
>
>     In addition, image generation tasks like depth-to-image controllable generation are similar to generative depth estimation, as both belong to per-pixel-level conditional generation tasks. Therefore, we also explore extensions to these types of image generation tasks. However, due to the constraints of the rebuttal format, we cannot show the visualization results (these tasks typically lack quantitative evaluation). We commit to presenting these visualization in the final version.
>
> 3. **To weakness 2: “Minor typo… .”**
>
>     **Reply**: typos: Plug-and-play -> play-and-plug
>
>
> [1] Show-o: One Single Transformer to Unify Multimodal Understanding and Generation, ICLR 2025.
>
> [2] A Simple Approach to Unifying Diffusion-based Conditional Generation, ICLR 2025.
>
> [3] One Diffusion to Generate Them All, CVPR 2025.

---

> > ### Author Response · Authors · 2025-08-05
> >
> > Dear Reviewer RUJQ,
> >
> > We sincerely appreciate your time and effort in reviewing our paper. We hope our explanations have addressed your concerns. As the discussion phase is nearing its end, we look forward to your reply. If further clarification is needed, please do not hesitate to mention it, and we will promptly address your inquiries. We look forward to receiving your feedback.
> >
> > Best regards,
> >
> > Paper 2750 Authors

---

> ### Author Response · Authors · 2025-08-07
>
> Dear Reviewer RUJQ,
>
> We sincerely thank your time and effort in reviewing our paper. We hope our explanations have addressed your concerns. As the discussion phase is nearing its end, please let us know if you have any further questions, and we would be happy to answer them. We look forward to receiving your feedback.
>
> Best regards,
>
> Paper 2750 Authors

---

### Note · Authors · 2025-08-12

We sincerely thank the AC and reviewers for their time and constructive feedback.

Before the rebuttal phase, our work received favorable feedback from R2 and R3, where they remarked, “Enabling powerful depth estimation without irreversibly degrading the inherent image generation capability of the pre-trained DiT is impressive; ****The proposed MERGE method … is simple and efficient.”.

During the rebuttal, we further clarify our work’s motivation and present additional experiments to address all concerns:

1. **Motivation**: Geometric estimation and image generation are generally considered as two separate tasks due to their distinct inherent paradigms. To explore how to enable geometric estimation without compromising the image generation capability of a pretrained T2I model, we present a new and simple paradigm that unifies geometric estimation and image generation within a single framework. Our method can effectively alleviate the development and deployment challenges caused by the growing number of specialized models for geometric estimation and image generation.
2. **Application scenarios**: The out-of-distribution (OOD) experimental settings demonstrate that our MERGE model has greater advantages in real-world application scenarios compared to other unified models. In addition, we extend MERGE to the normal estimation task, where it achieves overall better performance than fully fine-tuned, task-specific normal estimation models. This further highlights the advantages of our approach in real-world settings.
3. **Comprehensive comparison**: We add discussions comparing our approach with traditional depth estimation methods, conduct more detailed and fair ablation studies on training data, image generation quality evaluation, and provide additional Stable Diffusion versions of our MERGE model to show the superiority of our method.

After the rebuttal, **R2, R3, and R4 acknowledged the contributions of our method, reflected in their positive ratings.** In particular, Reviewer 4 confirmed, “**My concerns have been well addressed**,” and changed his rating from a negative score to a positive score. R1 may have been too busy to review our feedback. We sincerely hope that R1 will consider our response during the final stage and reconsider the rating.

---

### Decision · Program_Chairs · 2025-09-17

**Decision:**

Accept (poster)

**Comment:**

This paper proposes a framework for unifying image generation and depth estimation using pretrained text-to-image diffusion models. Instead of fine-tuning all parameters, the paper introduces lightweight plug-and-play converters alongside a Group Reuse mechanism to enable switching between modes while mitigating catastrophic forgetting. Experiments demonstrate strong depth estimation results across benchmarks while preserving image generation quality.

All four reviewers rated the paper as borderline accept. Reviewers RUJQ and Xt7v question the necessity of preserving image generation priors when depth-specific methods could be stronger. Generalization to tasks beyond depth remains unconvincing. Reviewer VXik was initially concerned about fairness and motivation but found the rebuttal satisfactory. Reviewer FL3u argues the paper is closer to a single-task method (depth estimation only) than a true unified model. Unlike methods such as OneDiffusion, MERGE does not extend naturally to a broader range of tasks. Moreover, compared to single-task baselines like Lotus, its depth results are not competitive.

After rebuttal, the reviewers converged on borderline accept, highlighting that the paper is technically sound. The rebuttal helped mitigate concerns regarding comparisons and generalization, though questions remain about whether MERGE should be considered a “unified” model given its limited scope relative to other frameworks. Given the positive consensus of the reviewers,  the AC does not oppose the acceptance. The authors are strongly requested to improve the paper according to the reviews. It is hoped that the work will stimulate further discussion in this direction.